# Urban Drool Water Quality in Denver, Colorado: Pollutant Occurrences and Sources in Dry-Weather Flows

**Forrest Gage Pilone** [1,2], **Pablo A. Garcia-Chevesich** [1,2,3] **and John E. McCray** [1,2,4,*]

1 Department of Civil and Environmental Engineering, Colorado School of Mines, Golden, CO 80401, USA; fgpilone@mines.edu (F.G.P.); pchevesich@mines.edu (P.A.G.-C.)
2 National Science Foundation Engineering Research Center, ReNUWIt, Golden, CO 80401, USA
3 Intergovernmental Hydrological Programme, UNESCO, 75007 Paris, France
4 Hydrologic Science and Engineering Program, Colorado School of Mines, Golden, CO 80401, USA
* Correspondence: jmccray@mines.edu

**Abstract:** Dry-weather flows in urban channels and streams, often termed "urban drool", represent an important source of urban surface water impairment, particularly in semi-arid environments. Urban drool is a combination of year-round flows in urban channels, natural streams, and storm-sewer systems (runoff from irrigation return flow, car washes, street cleaning, leakage of groundwater or wastewater into streams or storm sewers, etc.). The purpose of this study was to better understand the extent and sources of urban drool pollution in Denver, Colorado by identifying relationships between urban catchment characteristics and pollutants. Water-quality samples were taken throughout Denver at urban drainage points that were representative of a variety of urban characteristics. Samples were analyzed for total suspended solids (TSS), coliforms, *Escherichia Coli* (*E. coli*), nutrients (nitrate, phosphorus, and potassium), dissolved and total organic carbon, and dissolved and total recoverable metals. Results from this study were as follows: (1) most contaminants (nitrate, phosphorus, arsenic, iron, manganese, nickel, selenium, and zinc) were concluded to be primarily loaded from shallow groundwater; (2) anthropogenic effects likely exacerbated groundwater pollutant concentrations and contributions to surface water; (3) nitrate, nickel, and manganese may be partially contributed by industrial inputs; (4) medical marijuana cultivation sites were identified as a potential source of nutrient and zinc pollution; (5) *E. coli* was a ubiquitous contaminant in all urban waterways; (6) erosion of contaminated urban soils, presumably from construction, was found to significantly increase concentrations of TSS, total phosphorus, and total metals. Increasing urbanization and predicted drier climates suggest that dry-weather flows will become more important to manage; the results from this study provide insight on dry-weather water quality management for the City and County of Denver.

**Keywords:** dry-weather flows; urban drool; water quality; urban water management; urban characteristics

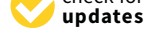

## 1. Introduction

Research on urban water impairment is abundant, but the degree of degradation and sources of pollution differ depending on the hydrological and water quality dynamics and specific characteristics of the urban area [1–3]. Notably, the majority of recent research on urban water impairment focuses on urban stormwater [4]. While stormwater runoff has a significant impact on urban pollutant loads, dry-weather urban flows in streams, channels, and storm sewers, termed "urban drool", has also been identified as an important source of contamination to urban waterways [4–9]. Dry-weather pollutant loads are especially relevant to consider in arid and semi-arid climates because less annual rainfall and longer dry periods result in more concentrated dry-weather contaminant loads being added to urban streams [4,5,7].

Many authors have conducted research on urban dry-weather pollution in countries such as Australia [2,10], Korea [11], France [12], and Singapore [13] among others. Additionally, significant research has been performed in the United States, where most research conducted in arid and semi-arid climates included California (e.g., References [4,6,14,15]) and Nevada (e.g., References [8,16,17]). Research results agree that consequences associated with urban water impairment justify the study of urban drool water quality. Nutrients have perhaps the most evident consequences through a process known as eutrophication, which is the degradation of water quality through excess nutrient input to waterways, where excess algae growth can deplete waterways of oxygen, impairing its use for fisheries, recreation, aesthetics, industry, irrigation, and drinking water [18,19]. Pathogens such as *Escherichia Coli* (*E. coli*) and total coliforms can cause human illness and prohibit drinking-water and recreational use [14,20]. Though different types of metals have different toxicities, most metal loading is considered degrading to urban streams and may have severe impacts on aquatic life. For example, selenium (Se) has been found to severely harm migratory waterfowl in impaired urban streams [21] and even small amounts of heavy metals such as copper (Cu), lead (Pb), and zinc (Zn) can have detrimental effects for benthic invertebrates [22], which are critically important for a healthy stream ecosystem. Dry-weather flows also contain more dissolved forms of pollutants than other nonpoint pollutant sources, which generally increases aquatic bioavailability and presents challenges when removing pollutants [15,16].

The South Platte River flows through Denver, Colorado and represents a source of recreation for people, habitat for aquatic life, and a supply of drinking water for Denver and downstream communities [23]. Similar to many urban waterways, urban processes and drainage within the city severely impair the river's water quality [24]. This metropolitan area is situated in a semi-arid climate and, similar to many urban waterways in the western United States, the South Platte River is dominated by wastewater effluent during periods of dry-weather [24]. However, wastewater treatment plants and other point sources have largely been characterized and are regulated under varying legislature such as the Clean Water Act (CWA) and surface water quality standards set by the Colorado Department of Public Health and Environment (CDPHE) [24–26]. Though these point sources of pollution certainly contribute to stream impairment, nonpoint sources have been identified as significant and uncontrollable sources of contamination and have the potential to produce pollutant loads at magnitudes equal to or greater than point sources [4,24,27–29]. Nonpoint sources can pollute wet-weather and dry-weather flows. While a couple studies have characterized Denver's wet-weather (stormwater) runoff quality (e.g., References [30,31]), a detailed investigation of dry-weather flows for Denver has not been published, evidencing a need for this study.

Because dry-weather flows represent a substantial source of urban water impairment, it is essential to understand how anthropogenic effects and characteristics of urban drainage under dry-weather conditions impair water quality. Achieving a better comprehension of dry-weather water quality can inform dry-weather pollution mitigation strategies, which differ from stormwater management strategies [9]. Additionally, climate change models predict that Denver and much of the western United States will become hotter and more arid [32], suggesting that urban dry-weather flow will become a more significant resource as water demand and scarcity increase. Considering all this, the purpose of this study was to characterize the extent and potential sources of pollutants in urban drool in Denver to enable potential mitigation strategies.

## 2. Background Information

### 2.1. Urban Drool: Dry-Weather Urban Water Quality

Dry-weather flows, or urban drool, derives primarily from "nuisance" flows (e.g., over-irrigation of ornamental landscaping and turf, car washes, street cleaning, irrigation return flows), permitted and illicit point sources of discharge (e.g., discharges from industrial, construction, and dewatering operations), domestic wastewater and drinking water infras-

tructure leaks or overflow, "natural" non-urban water flowing from outlying areas, and groundwater inflows [8,9,11,12,16,17,24,33,34]. The most predominant sources of nonpoint dry-weather flow in semi-arid climates include irrigation return flow and shallow groundwater contributions. Irrigation return flow is a nuisance flow and most often comes from residential, open space, and agricultural land uses where excess irrigation of turf grass, ornamental plants, and other vegetation create excess runoff, which can contribute nutrient and metal pollution [8,27]. Groundwater flow is an important and prominent source of dry-weather urban flows. Groundwater contributions come from a variety of sources, including, but not limited to, construction dewatering, weep holes (breaks in storm sewers where groundwater seeps), natural seeps, and instream contributions [33].

The City and County of Denver Department of Public Health and Environment (DDPHE) conducts regular studies on urban surface water throughout the city and inquiries storm drain outfalls along the South Platte River in order to assess and inform water quality decisions [23]. Despite their efforts, the DDPHE reported in its 2020 water quality report for small urban tributaries to the South Platte River that water quality is mostly impaired and does not appear to be improving [35]. Identified pollutants of concern were *E. coli*, nitrogen (N), phosphorus (P), and Se [35]. Though this sampling program highlights important water quality data for the City and County of Denver, it neglects to specifically focus on dry-weather water quality, analyze dry-weather flows from outfalls, or to identify sources (i.e., understand relationships between urban characteristics and pollutant types to understand where pollutant sources may be derived), illustrating data gaps that need to be filled to obtain a useful understanding of the dynamics of dry-weather urban water quality in Denver.

In addition to the data provided by the DDPHE, studies from other arid/semi-arid urban areas found that pollutants predominant in dry-weather flow include Se [8,15], manganese (Mn) [8], N [2,6,10,17], P [2,12,17], total coliforms [8,14,36], and *E. coli* [9,28]. Studies that analyzed both wet and dry-weather flows found that metal loads (with the exception of Se and Mn) were significantly greater during wet-weather flows [2,6,11,37]. However, metals are still present at concerning concentrations in dry-weather loads, as evidenced by several investigations [7,9,27].

### 2.2. Effects of Urbanizaion and Land Use Characterisitcs

Urbanization is the most clear and evidenced factor that leads to impaired dry-weather water quality. The urban environment severely complicates natural hydrologic cycles and introduces new avenues for pollutants to enter waterways [33,38–40]. Urbanization includes covering of pervious areas, vegetation clearing, soil compaction, introduction of artificial drainage, implementation of water infrastructure, and significant alteration of catchment topography, which all affects water quality and quantity in the urban environment [33,38]. Urbanization has been found to increase temperature, P, N, total suspended solids (TSS), and pathogens (total coliforms and *E. coli*) in urban drool [41–44]. A study by Gustafson et al. [30] found that most likely because of increased imperviousness associated with urbanization, certain watersheds in Denver experienced greater nutrient and metal loads.

Notably, the addition of hydrological connections, such as gutters, drains, storm sewers, and other impervious conveyances create direct connections from polluting urban activities to receiving water bodies with minimal retardation, chemical or biological reaction, or filtration, significantly affecting pollutant loads as well as altering the natural hydrologic cycle [38,43,44]. These hydrologic connections increase the impacts of nonpoint pollution sources by exposing dry-weather flows to street litter, fossil fuel combustion, eroded vehicle components, corroding galvanized building materials, waste (pet, human), and fertilizers/pesticides [9,41]. For example, several studies found that increased P in urban surface water may be due to greater impervious connections from sources of P to receiving bodies of water [45,46]. This principle may explain why high density residen-

tial and urban land uses have been found to contribute more pollutants than other land uses [47].

While urbanization may have obvious changes on the surface, it also significantly affects the subsurface of the urban environment, which is important in the context of this investigation since groundwater contributions are a significant source to dry-weather flows. Effects include, but are not limited to, use of foreign soil, soil compaction, increased shallow groundwater recharge, and subsurface infrastructure [1,33,48–50].

Specific land uses and land cover characteristics further reveal what exactly about these urban areas are contributing pollution. A number of studies from similarly semi-arid urban areas found that metal pollution was predominantly loaded through industrial and commercial land use inputs, where industrial emissions may especially be of importance during dry-weather [51–53]. Conversely, recreational sites, such as parks and open space land uses, were found to have greater contributions of bacteria such as *E. coli* [52]. Land use also affected the type and form of pollutants, where land uses with greater vegetation densities had more particulate based pollutants, and higher density urban land uses had more dissolved pollutant quantities [54].

Shallow groundwater is also affected by varying land use. A notable anthropogenic pollutant in groundwater is nitrate, where industrial and agricultural land uses have been reported to increase groundwater nitrate concentrations [55–57]. Anthropogenic manipulation of groundwater associated with urban and agricultural land uses can also have severe effects on naturally-occurring pollutant concentrations in groundwater, such as arsenic (As) and Se [57,58]. Historic land uses prior to urbanization, primarily agriculture and industry in Denver, may be a strong contributor to legacy pollution in a watershed and may be a source of nutrients [59–61] and heavy metals [53], respectively. Pollutants inputted from past land uses can persist in the subsurface for many decades and eventually discharge to surface water several years later, depending on the transport mechanisms of the pollutants and the hydraulic residence time of the groundwater [62]. In addition, land use changes can affect local hydrologic, geochemical, and redox conditions which may encourage previously sequestered pollutants to mobilize and contaminate surface water [63].

## 3. Materials and Methods

### 3.1. Study Area and Site Selection

Data on outfall flow observations were used to locate drainage points along the South Platte River and was provided by the City and County of Denver's Department of Transportation and Infrastructure (DOTI). Points that had reported dry-weather flow were noted and were visited in the field to observe accessibility, safety, flow rate, type of drainage, and other relevant notes concerning sampling and land use.

To refine potential sampling locations, the points were displayed overlain with Denver's storm sewer mains and Earth imagery on ArcGIS Desktop (ESRI 2019, ArcGIS Desktop: Release 10, Environmental Systems Research Institute, Redlands, CA, USA) to visually understand the size and nature of the catchment area expected to drain to each point. Sampling locations were selected based on continuity of year-round dry-weather flows, and inclusion of a variety of urban land-uses and different urban landscapes. For land uses, we desired a range of sampling locations that included industrial land use and high imperviousness, to residential and open space land uses and lower imperviousness. In addition, a "background" site was selected in Golden, Colorado, a suburb roughly 15 miles west of Denver, that comprised of mostly low-density residential and commercial land uses. If sampling sites did not have a designated name, a nearby landmark or location was used in its place. A total of ten sampling sites were selected. The general characteristics of each site is shown in Table 1. Each site had a variety of drainage types, evidencing different drainage water quality and sources of dry-weather flow. The locations and watershed boundaries of sampling sites are displayed in Figure 1.

**Table 1.** Basic sampling site descriptions.

| Sampling Site Name | Site ID | Drainage Outlet Type | Channel Bottom Type |
|---|---|---|---|
| South Golden Gulch | SG | Gulch | Natural/Concrete |
| River North Art District | RN | Storm Drain Outfall | Concrete |
| Five Points | 5P | Storm Drain Outfall | Concrete |
| Denver Skate Park | D8 | Storm Drain Outfall | Concrete |
| Cherry Creek | CK | Gulch | Natural/Concrete |
| Lakewood Gulch | LG | Gulch | Natural |
| Weir Gulch | WG | Gulch | Natural/Concrete |
| Denver Wastewater Building | DW | Storm Drain Outfall | Concrete |
| Sanderson Gulch | SD | Gulch | Natural |
| West Harvard Gulch | WH | Gulch | Natural/Concrete |

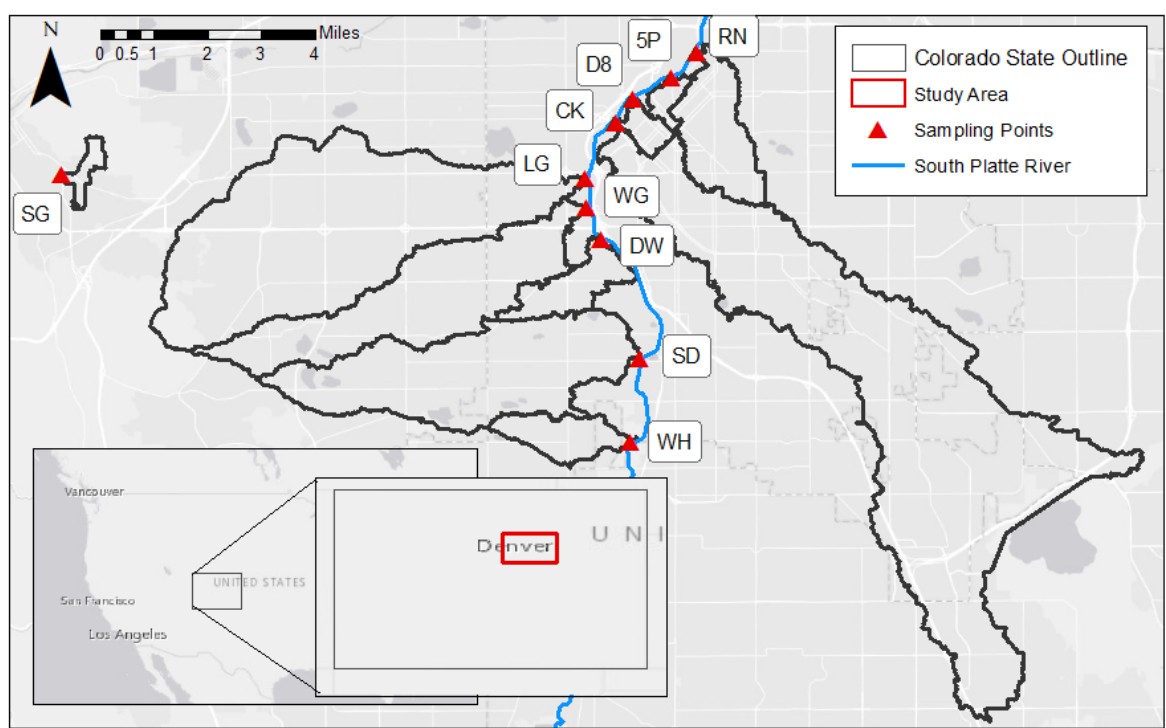

**Figure 1.** Study area with sampling locations and their respective drainage areas in Denver and Golden, Colorado.

### 3.2. Watershed Delineation and Characterization

An in-depth GIS analysis was performed to determine characteristics for the catchment of each sampling location, including watershed size, land cover, land uses, impervious surface, and tree canopy cover. The GIS platform used was ArcMap version 10.7.1. To obtain an urban watershed indicative of all probable flow paths to the drainage point, the City and County of Denver's and City of Golden's storm sewer main pipe networks were "burned" into the digital elevation model (DEM) by subtracting approximate pipe depths from the DEM. Environmental Systems Research Institute (ESRI) Spatial Analyst Toolbox and its Hydrology Tools were used to delineate urban watersheds of each sampling point, as seen in Figure 1 above.

Land cover data were used from two different, reputable sources. The first was the National Land Cover Database (NLCD) for land cover in the year 2019 (NLCD19) [64]. NLCD19 was the most up to date land cover dataset available. NLCD19 is a product from the Multi-Resolution Land Characteristics Consortium (MRLC). The second land cover dataset used was the Denver Regional Council of Governments (DRCOG) Area High Resolution Land Cover [65]. This land cover layer was specific to the Denver metropolitan area. The DRCOG dataset was created using the U.S. Department of Agriculture (USDA)

one-meter-resolution National Agriculture Imagery Program (NAIP) aerial imagery from 2017. Planimetric datasets were also used to reflect building footprints, sidewalks and driveways, roads, and parking lots. Each layer was projected, along with the delineated catchments, and clipped according to each delineated basin's outline. Both land cover layers were raster datasets, so relative percentages of each land cover class were reported by taking each land cover's raster count divided by the total raster count in the basin.

Though the DRCOG land cover database included impervious surface as a land cover class, more detailed datasets existed that more accurately define imperviousness. Notably, the NLCD19 had a raster dataset specifically for percent developed imperviousness [64]. Within this dataset, each raster cell was assigned a value 0–100, which designated the degree of imperviousness. Tree Canopy was another available NLCD dataset, though this layer was taken from the 2016 NLCD package [66]. The tree canopy dataset also assigned each raster cell a value 0–100, which represented the percent of tree canopy cover. After projecting and clipping these layers to each drainage basin, a weighted average was taken to determine the percent imperviousness and tree canopy values for each watershed.

The City and County of Denver produced a specific land use dataset for Denver's 2018 zoned land uses, which was the city's most recent available assessment. These data were accessed from the City and County of Denver's Open Data Catalog [67]. The dataset was compiled from parcel records, which generalized their land-use classification into specified categories. However, this dataset only included parcels within Denver County. Several chosen catchments overlapped to other counties, including Jefferson County to the west of Denver and Arapahoe County to the southeast. Specific land use layers for these counties were not available; however, parcel data were available on each county's website that included specific descriptions of the parcel [68,69]. These parcel descriptions were manually sorted to match with the urban land use categories used by the City and County of Denver. Because these layers were raster datasets, the same method as land cover analysis was used to determine the percentages of land use in each catchment.

Another relevant land use that was considered in this work were the locations of medical marijuana cultivation (MED) sites. Large-scale cultivations are required to properly dispose of their waste, but smaller (personal) MED sites are not highly regulated, and preliminary data suggest that these sites may be contributing to nutrient pollution in the urban environment (discussed later). Approved MED licenses are publicly searchable and included the addresses of MED sites within the Denver Metropolitan Area. These data are available from Colorado Department of Revenue (CDOR) Marijuana Enforcement Division [70]. Addresses were converted to coordinates using the online application GPS Visualizer (Portland, OR, USA) [71]. The MED sites were then overlaid across the delineated catchment areas and the number of MED sites in each basin was determined.

### 3.3. Field Collection

Dry-weather samples were collected after waiting a minimum of three days of dry-weather conditions. This was determined using a precipitation reporting application from Community Collaborative Rain, Hail, and Snow Network (CoCoRaHS, Fort Collins, CO, USA), where total daily precipitation was monitored and recorded for the Denver metropolitan area [72]. CoCoRaHS included at least one monitoring station in each catchment with the exception of one (SG). In this case, an adjacent monitoring station was used. If only trace amounts or no detections of precipitation were recoded, it was considered dry-weather conditions. By design, sampling was often conducted after more than three-days of dry weather in order to gather data to determine if pollutants were correlated with the number of antecedent dry-days.

A total of 16 sets of samples were collected from each site between July 2020 and June 2021. Samples were collected at least once a month (samples taken at least once a month are considered sufficient to characterize a watershed [73]) and not more than twice a month. However, the months of January and March had no samples due to persistent snow, ice, and/or rain that prevented design collection conditions. Grab samples were taken

using 500 mL amber glass bottles. Water was collected as near to the center of the stream as possible. Bottles and bottle caps were rinsed with urban drool several times before filling with minimal headspace. After collection, bottles were sealed and immediately placed on ice, recording the time of collection. During sampling, pH, conductivity, and temperature were measured in-stream with an Orapxi multifunction water quality meter, model number EZ-9908.

Flow rate estimates were taken at each site. If available, U.S. Geological Survey (USGS) flow data were used for streamflow from the nearest stream gage. One site (DW) was equipped with a v-notched (triangular) weir. The Kindsvater–Shen equation was used to determine flow for the v-notched weir, which is standardized by the United States Bureau of Reclamation [74]. However, most of the drainage points were not equipped with a USGS gaging station or a weir, so manual methods of streamflow measurement had to be applied. Dry-weather flow is relatively small, so using a traditional stream flow device was not feasible at most sites. Due to this restriction, a modified version of the float method [75] was utilized for the majority of sites to estimate stream flow rates. This method estimates surface velocity with the use of a floating object. Because surface velocity is usually higher than actual stream velocity, observed velocity data were multiplied by 0.78 to better estimate the actual velocity [75] before calculating flowrate. The cross-sectional area was calculated with the average width and average depth of the stream along the distance that the float was measured. The velocity and area were then multiplied to obtain streamflow. While efficient and accurate measurement of low flows in urban channels is known to be difficult, this method provided a consistent basis to compare streamflow for use in our data analysis.

### 3.4. Laboratory Analysis

After field collection, laboratory analysis was conducted either immediately or the next day. The maximum hold time differed for each analysis and is reported in Table 2, along with standard methods used for analysis. Pollutant concentration data were utilized in place of load calculations since CDPHE standards and DDPHE data were reported as concentrations, which allowed for direct comparisons to be made.

**Table 2.** Laboratory analysis methods, standards, and/or equipment and hold time.

| Analysis | Method/Standard/Equipment | Maximum Sample Hold Time |
|---|---|---|
| Total Coliforms | Idexx Colilert | 24 h |
| *E. coli* | Idexx Colilert | 24 h |
| Nitrite | HACH TNT 839 | 24 h |
| Nitrate | HACH TNT 835 | 24 h |
| Ammonia | HACH TNT 831 | 24 h |
| Phosphate | HACH TNT 843 | 24 h |
| TSS | EPA Standard Method 2540D | 7 days |
| DOC/TOC | Shimadzu TOCV-TNM-LCSH | 24 h |
| Total Nitrogen | Shimadzu TOCV-TNM-LCSH | 24 h |
| Total Dissolved Metals | ICP-AES | 7 days |
| Total Recoverable Metals | ICP-AES | 7 days |

ICP-AES = inductively coupled plasma atomic emission spectroscopy.

### 3.5. Water Quality Standards

Water quality standards are set by federal, state, and local governments to regulate and control the concentration and/or mass of pollutants in waterways. The most relevant standards for this study was CDPHE Regulation Number 38 (Reg. 38), which specifies concentration standards for specific segments of the South Platte River Basin [76]. In this study, we used the Reg. 38 standards for the South Platte River segment that runs through the length of Denver for pH, *E. coli*, nitrate, As, cadmium (Cd), Cu, iron (Fe), Pb, Mn, nickel (Ni), Se, and Zn standards. Several of these standards were table value standards (TVS), which are standards dependent on local water parameters including pH, temperature, and

hardness. Average values from the South Platte River prior to entering Denver County were used for TVS calculations. CDPHE Regulation Number 85 (Reg. 85) was another regulation considered for P [77].

It is important to note that these CDPHE standards are in-stream concentration standards and do not regulate nonpoint dry-weather urban flows. However, these standards were used because our drainages discharge to the South Platte River, and it is beneficial to compare drainage water quality to in-stream standards.

### 3.6. Statisitcal Analysis

Samples that were below the detection limit (BDL) may be any value between zero and the analyte's detection limit (DL). In order to account for these values in the analysis, samples that were BDL were assigned a value of one half of the DL because values of zero do not allow for computation of geometric statistics and use of the DL may artificially increase the analyte's statistics [52]. Statistical analyses were performed at 95% confidence level ($p < 0.05$).

General and descriptive statistics were used to portray basic statistical data and were produced using built-in Excel functions. Graphical displays of this data included boxplots and scatter plots, which were created using the open-source R data visualization package ggplot2 from Tidyverse [78]. The Shapiro–Wilk normality test was used to evaluate how good each variable data fit within a normal probability distribution. The purpose of this test was to inform what statistical analyses were applicable since parametric analyses lose power when data are not normally distributed [79]. The built-in R command function Shapiro.test was used.

The purpose of the one-factor premutation test for difference in means (one-factor permutation test) was to determine if statistically significant differences existed between the means of the independent groups of data, where the independent groups were the sampling site locations. In many applications, analysis of variance (ANOVA) is used for this analysis; however, since much of the data appeared to be skewed or non-normal according to the Shapiro–Wilk test for normality, the one-factor permutation test was used. Hirsch et al. [79] reported that the one-factor permutation test produces very similar results as ANOVA, but is not sensitive to non-normal data distributions. The test was computed using the permKS command from the perm package [80] in R. Another more "user-friendly" R function was used to confirm the permKS results, which was the perm1way R command taken from the Supplementary Materials from Hirsch et al. [81]. ANOVA analysis was also conducted for comparison as well.

The one-factor permutation test was not able to indicate which group(s) specifically were significantly different. Thus, another test was necessary in order to distinguish significant differences between specific groups. The most common method to determine whether two groups are significantly different is the *t*-test. However, the parametric *t*-test requires that both groups follow a normal distribution and that the groups have equal variances. As a result, the two-sample permutation test for difference in means (two-sample permutation test) was used to account for differences in variance and data distributions among groups and variables [79]. This test was computed using the perm2 function that was provided in the Supplementary Materials from Hirsch et al. [81].

Correlation analysis measured the strength of association between two variables. Correlation analysis was used to measure the strength of relationships between urban characteristics and mean pollutant concentrations of the sampling sites. A confidence interval of 90% was also considered ($p < 0.10$) to identify weaker relationships that could likely still be meaningful. Only the pollutants of concern as identified through literature review and those that exceeded CDPHE standards were analyzed to the urban characteristics. Pearson's r and Spearman's rho correlations are most often used in correlation analysis. Spearman's rho correlation was preferred though since Pearson's r correlation identifies only linear relationships, whereas Spearman's rho measures monotonic relation-

ships, which is able to identify non-linear relationships as well [79]. The cor.test function in R was used to calculate Spearman's rho and its associated significance.

## 4. Results

### 4.1. GIS Analysis

The delineated catchments for the sampling points within the Denver Metropolitan Area are illustrated in Figure 1, while Table 3 shows the percent of each urban characteristic present in the watershed. The variations in size, land cover, land use, imperviousness, and tree canopy cover translate into distinct urban characteristics.

The most urban/developed watersheds were those with higher percentages of imperviousness, medium- and high-intensity development, multi-unit residential land use, office land use, and industrial land use. These sites were RN, 5P, D8, CK and DW (see Figure 1). Less developed sites consisted of lower percentages of imperviousness and higher percentages of open space and low-intensity development, tree canopy coverage, herbaceous land cover, turf/irrigated surface land cover, and single-unit residential land use. These sites were SG, LG, WG, SD, and WH. Some notable observations from this analysis include the following: (1) SG was the smallest catchment and had the largest proportion of open space and suburban land uses; (2) RN, 5P, D8, and DW were among the most impervious watersheds and also were storm drain outfalls; (3) CK, LG, WG, and SD were large surface gulches that drained a variety of different urban characteristics, starting in outlying residential areas of Denver traveling through progressively more urban land uses; and (4) 5P, DW, and WH drained significant portions of industrial land uses.

**Table 3.** Percentages of urban characteristics for each site. Sources for each characteristic are denoted in table footer.

| Urban Characteristic | SG | RN | 5P | D8 | CK | LG | WG | DW | SD | WH |
|---|---|---|---|---|---|---|---|---|---|---|
| Watershed Area (km$^2$) | 1.06 | 7.69 | 1.11 | 1.42 | 66.48 | 43.07 | 20.72 | 4.25 | 23.96 | 3.96 |
| Development, Open Space [1] | 10.7% | 3.6% | 0.7% | 0.4% | 13.4% | 12.2% | 9.5% | 1.7% | 6.4% | 7.3% |
| Development, Low-Intensity [1] | 14.0% | 19.5% | 0.5% | 0.8% | 33.2% | 37.6% | 44.5% | 29.0% | 48.4% | 54.7% |
| Development, Medium-Intensity [1] | 23.5% | 48.4% | 4.8% | 15.5% | 32.6% | 28.1% | 33.0% | 40.4% | 32.2% | 26.1% |
| Development, High-Intensity [1] | 11.9% | 28.5% | 94.0% | 83.4% | 18.1% | 12.9% | 8.1% | 28.8% | 7.2% | 11.5% |
| Herbaceous [1] | 28.3% | 0% | 0% | 0% | 0.6% | 5.0% | 1.6% | 0% | 0.4% | 0.2% |
| Imperviousness [1] | 31.3% | 64.2% | 90.3% | 87.6% | 50.0% | 43.8% | 44.7% | 62.4% | 44.7% | 44.7% |
| Tree Canopy Cover [1] | 4.0% | 2.3% | 0.2% | 0.1% | 6.8% | 6.0% | 4.1% | 1.3% | 3.0% | 1.9% |
| Turf/Irrigated Land [2] | 8.7% | 12.4% | 1.2% | 1.2% | 20.0% | 20.9% | 23.8% | 18.6% | 25.2% | 27.3% |
| Vacant [3] | 34.5% | 3.9% | 23.7% | 7.3% | 3.3% | 5.9% | 3.7% | 3.9% | 5.0% | 3.3% |
| Multi-Unit Residential [3] | 8.3% | 23.5% | 14.3% | 14.3% | 18.2% | 7.7% | 5.7% | 8.1% | 12.4% | 2.7% |
| Single-Unit Residential [3] | 18.0% | 27.0% | 0% | 0.1% | 35.4% | 39.2% | 54.0% | 45.5% | 52.1% | 68.2% |
| Industrial [3] | 0.5% | 5.1% | 14.2% | 1.9% | 2.5% | 2.8% | 1.3% | 19.3% | 0.5% | 10.0% |
| Office [3] | 0.3% | 6.1% | 6.1% | 24.8% | 6.2% | 9.6% | 1.3% | 1.2% | 0.6% | 1.2% |
| Park/Open Space [3] | 9.7% | 7.4% | 0.1% | 3.9% | 13.0% | 3.4% | 4.8% | 2.5% | 5.2% | 2.2% |
| Commercial/Retail [3] | 9.8% | 6.6% | 5.7% | 7.5% | 6.6% | 5.0% | 4.7% | 7.8% | 5.7% | 1.7% |
| MED Sites [4] | 0 | 29 | 4 | 12 | 94 | 17 | 10 | 24 | 11 | 2 |
| MED Density [4] (site/km$^2$) | 0 | 3.77 | 3.60 | 8.45 | 1.41 | 0.39 | 0.48 | 5.65 | 0.46 | 0.51 |

[1] NLCD19, [2] DRCOG, [3] Denver, Jefferson, and Arapahoe County Land Use, [4] CDOR Marijuana Enforcement Division. Note that several characteristics were omitted due to irrelevance or redundancy.

### 4.2. Dry-Weather Urban Flow Analysis

Boxplots are shown for contaminants that exceeded CDPHE standards in Figures 2–4. These standards were denoted by a dashed horizontal line in the boxplot. Boxplots illustrate the median, interquartile range, minimum and maximum values, and outliers in the data. The one-factor permutation test results are located in Table S1 in the Supplementary Materials and show which pollutants were statistically different among sites.

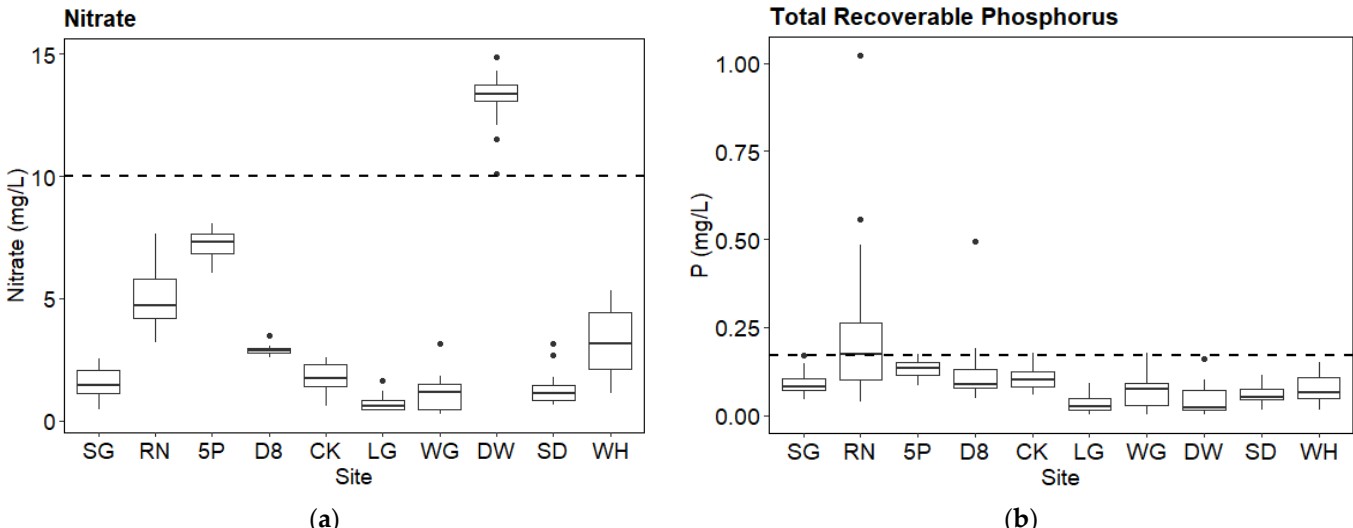

**Figure 2.** Nutrients that exceeded CDPHE standards were nitrate and total P: (**a**) Boxplot results with the acute CDPHE standard for nitrate. (**b**) Boxplot results with the CDPHE standard for total P.

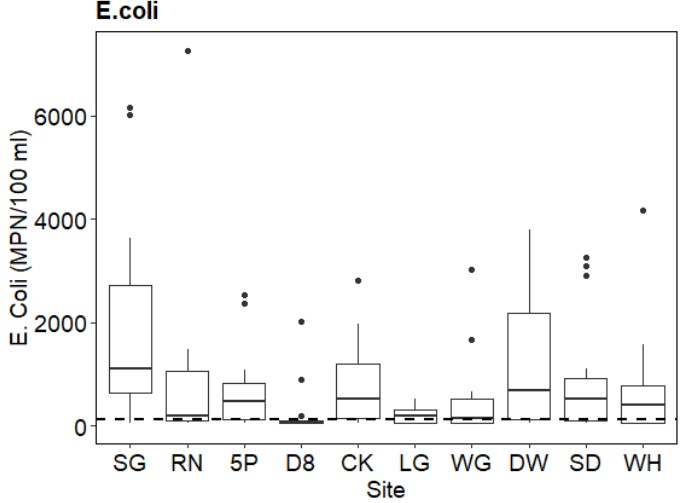

**Figure 3.** Boxplot results with the CDPHE standard for *E. coli*.

4.2.1. Flow

Table 4 shows flow data collected from each site, including flow method, flowrate in cubic feet per second (CFS), and standard deviation with relative standard deviation (RSD). The largest flow was observed at CK. Surface gulches (SG, CK, LG, WG, SD, WH) had larger flowrates than storm drain outfalls (RN, 5P, D8, DW). In addition, surface gulch watershed size appeared to be positively correlated with flowrate, whereas storm drain watershed size was not correlated to flowrate.

**Table 4.** Mean flowrates and standard deviation for each site along with measurement methods.

|  | SG | RN | 5P | D8 | CK | LG | WG | DW | SD | WH |
|---|---|---|---|---|---|---|---|---|---|---|
| Flow Measurement | Float | Float | Float | Float | USGS Gage | USGS Gage/Float | USGS Gage/Float | V-Notch Weir | Float | Float |
| Mean Flowrate (CFS) | 0.031 | 0.109 | 0.286 | 0.876 | 19.6 | 4.71 | 1.39 | 0.342 | 1.45 | 0.089 |
| Standard Deviation (RSD) | 0.015 (48.3%) | 0.058 (52.9%) | 0.052 (18.2%) | 0.204 (23.4%) | 16.7 (85.1%) | 3.12 (66.3%) | 1.40 (100.7%) | 0.506 (147.8%) | 1.35 (40.9%) | 0.087 (97.9%) |

LG and WG USGS gages were seasonal, so float method was utilized when gages were inactive.

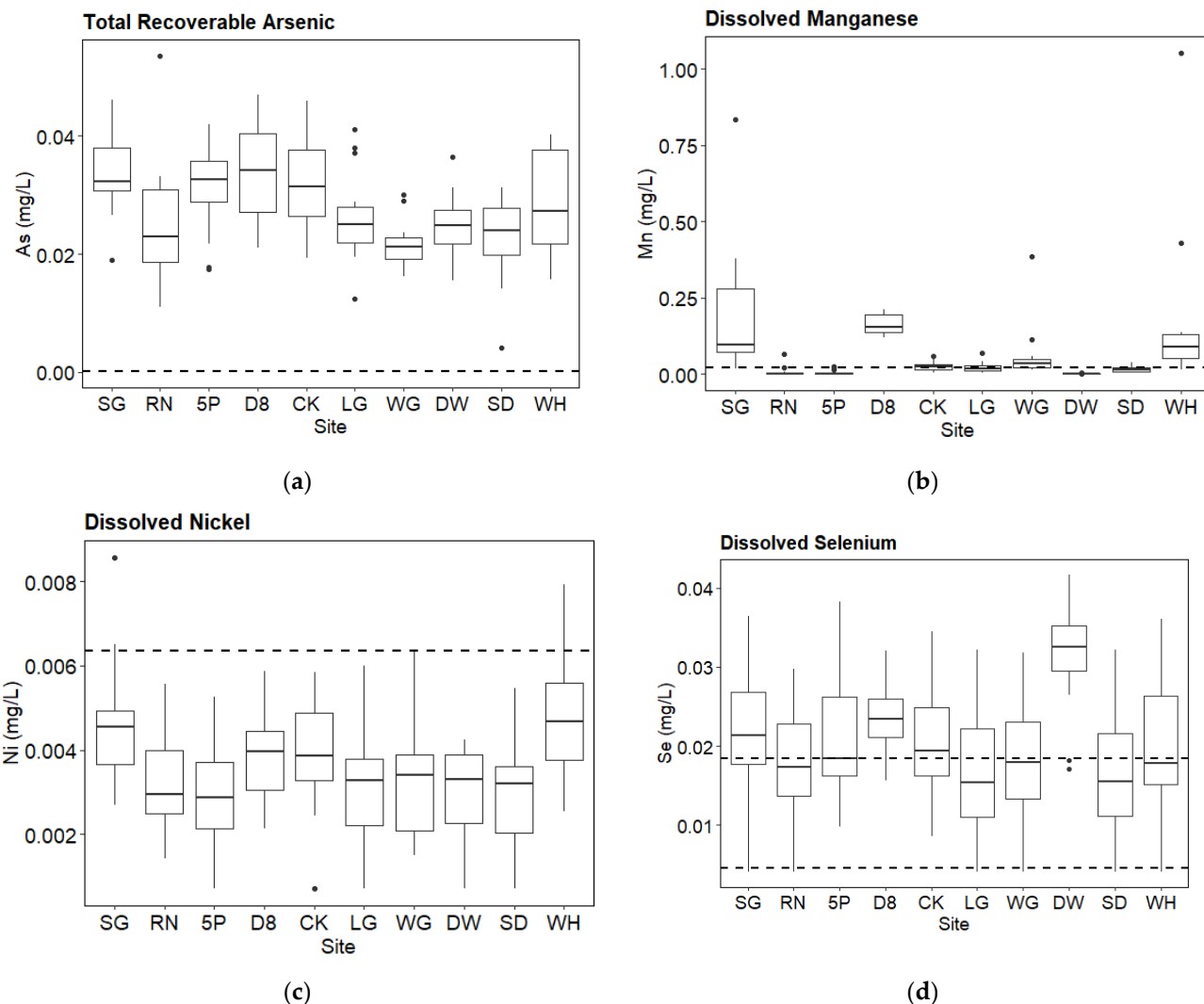

**Figure 4.** Metals that exceeded CDPHE standards were total As, dissolved Mn, dissolved Ni, and dissolved Se: (**a**) Boxplot results with the chronic CDPHE standard for total As. (**b**) Boxplot results with the chronic CDPHE standard for dissolved Mn. (**c**) Boxplot results with the chronic CDPHE standard for dissolved Ni. (**d**) Boxplot results with the acute and chronic CDPHE standards (acute standard was higher than chronic) for dissolved Se.

4.2.2. Nutrients

Table 5 shows the mean concentration values for nutrients from each site along with its standard deviation. Figure 2 shows boxplots for nitrate and total P. This figure shows that exceedances of CDPHE standards by nitrate and total P. Nitrite and ammonia were below CDPHE standards.

Nitrate concentrations were significantly different among sites as determined by one-factor permutation test. DW exceeded CDPHE acute nitrate standards upon every sample analysis, whereas other sites were consistently compliant. Further, 5P, RN, WH, and D8, experienced significantly greater concentrations than CK, SD, SG and LG based off the two-sample permutation test. Total nitrogen and nitrate concentrations were very similar, suggesting that nitrate was the predominant form of N-species in dry-weather flows.

Dissolved and total P were also significantly different among sites. The RN site exceeded the CDPHE total P standard about half of the time. RN had a significantly greater concentration of dissolved and total P among sites, and was followed by SG, 5P, D8, and CK as determined by two-sample permutation analysis.

Potassium (K) did not have any standards to compare to, but two-sample permutation analysis found that K was significantly greater at RN, 5P, D8, and CK than all other sites.

**Table 5.** Mean nutrient concentration values and standard deviation for each site. Results are reported in mg/L.

| | SG | RN | 5P | D8 | CK | LG | WG | DW | SD | WH |
|---|---|---|---|---|---|---|---|---|---|---|
| Nitrate | 1.51 (±0.63) | 5.04 (±1.29) | 7.20 (±0.60) | 2.87 (±0.21) | 1.73 (±0.65) | 0.71 (±0.34) | 1.12 (±0.76) | 13.2 (±1.14) | 1.31 (±0.69) | 3.33 (±1.36) |
| Nitrite | 0.02 (±0.01) | 0.03 (±0.04) | 0.02 (±0.03) | 0.02 (±0.04) | 0.01 (±.01) | 0.003 (±0.01) | 0.04 (±0.03) | 0.02 (±0,01) | 0.01 (±0.02) | 0.07 (±0.02) |
| Total Nitrogen | 1.65 (±0.65) | 5.29 (±1.30) | 8.05 (±1.11) | 3.22 (±0.38) | 2.00 (±0.63) | 0.86 (±0.36) | 1.55 (±0.87) | 14.0 (±1.56) | 1.6 (±0.66) | 3.59 (±1.32) |
| Ammonia | 0.03 (±0.02) | 0.07 (±0.09) | 0.03 (±0.04) | 0.03 (±0.04) | 0.02 (±0.02) | 0.03 (±0.02) | 0.02 (±0.01) | 0.12 (0.16) | 0.06 (±0.15) | 0.04 (±0.03) | 0.06 (±0.03) |
| P | 0.08 (±0.03) | 0.21 (±0.23) | 0.14 (±0.03) | 0.13 (±0.10) | 0.10 (±0.04) | 0.03 (±0.03) | 0.06 (±0.05) | 0.05 (±0.04) | 0.03 (±0.02) | 0.06 (±0.03) |
| P (T) | 0.09 (±0.03) | 0.25 (±0.15) | 0.13 (±0.03) | 0.13 (±0.10) | 0.11 (±0.04) | 0.04 (±0.03) | 0.07 (±0.05) | 0.04 (±0.04) | 0.06 (±0.03) | 0.08 (±0.04) |
| K | 2.37 (±0.25) | 7.92 (±6.62) | 7.33 (±0.94) | 10.9 (±2.17) | 7.23 (±0.84) | 2.13 (±0.49) | 2.86 (±0.80) | 1.80 (±0.36) | 2.56 (±0.60) | 3.17 (±0.99) |

(T) denotes total concentration. All other analytes are dissolved.

#### 4.2.3. Pathogens and Physical Parameters

Table 6 shows mean concentration values and standard deviation for pathogens and notable physical parameters, including TSS and total organic carbon (TOC) from each site. Figure 3 shows the exceedance of CDPHE standards for *E. coli* at every analyzed site. Constant exceedances suggested that *E. coli* contamination was ubiquitous in urban drool. High variations of pathogens are also evidenced by high standard deviations and large interquartile ranges in Table 6 and Figure 3, respectively.

*E. coli* and total coliforms were not significantly different among sites according to the one-factor permutation test due to the high variability in the data. TSS was also not significantly different among sites; however, it did appear that surface streams had greater TSS than storm drains. TOC was different among sites where storm drains, with the exception of RN, had significantly lower concentrations of TOC than surface streams as indicated by two-sample permutation tests.

**Table 6.** Mean *E. coli*, total coliforms, TSS, and TOC concentration values for each site along with their standard deviation. *E. coli* and total coliforms are reported in most probable number per 100 mL sample (MPN/100 mL), and TSS and TOC are reported in mg/L.

| | SG | RN | 5P | D8 | CK | LG | WG | DW | SD | WH |
|---|---|---|---|---|---|---|---|---|---|---|
| *E. coli* | 2138 (±1956) | 1248 (±2066) | 869 (±834) | 440 (±693) | 905 (±836) | 295 (±152) | 743 (±918) | 5458 (±14,850) | 1115 (±1224) | 937 (±1159) |
| Total Coliforms | 15,992 (±21,045) | 17,756 (±32,881) | 13,425 (±8435) | 6089 (±10,998) | 4911 (±885) | 5501 (±4706) | 8763 (±8215) | 25,827 (±35,200) | 11,360 (±11,108) | 13,704 (±20,370) |
| TSS | 1.50 (±2.50) | 66.7 (±219) | 0.79 (±1.78) | 0.50 (±1.24) | 10.2 (±5.65) | 5.01 (±5.51) | 8.61 (±5.71) | 0.54 (±1.33) | 9.64 (±15.7) | 6.58 (±4.04) |
| TOC | 4.32 (±0.85) | 5.85 (±3.75) | 2.44 (±0.96) | 2.51 (±1.03) | 4.30 (±0.84) | 3.36 (±0.92) | 6.30 (±3.44) | 3.30 (±0.27) | 5.11 (±0.99) | 7.35 (±7.87) |

#### 4.2.4. Metals

Table 7 displays mean concentration values of dissolved and total metals found in urban drool flows at each site along with their standard deviation. Notably, Cd, Cu, Cr, and Pb were often BDL. For this reason, these pollutants are not of particular concern in dry-weather urban flows and further analysis was not performed on these pollutants.

Figure 4 shows boxplots for metal contaminants that exceeded CDPHE standards, which included As, Mn, Ni, and Se. Chronic standards were often exceeded, but the acute standard for dissolved Se was exceeded, suggesting that Se may be a metal of particular concern. The one-factor permutation test found that both dissolved and total forms of As, Mn, Ni, and Se were significantly different among sites in addition to dissolved Fe and total Zn. Two-sample permutation analysis concluded the following: (1) dissolved and total As was greater at SG, 5P, D8, CK, and WH, where the significantly greatest concentration was found at D8; (2) dissolved Fe was significantly greater at SG, RN, WG, and WH; (3) SG, D8, and WH had significantly greater concentrations of dissolved and total Mn than the rest of the sites; (4) WH and SG had significantly greater concentrations of dissolved and total Ni compared to the other sites; (5) DW and D8 had significantly higher concentrations of dissolved Se, and DW, D8, SG, and WH had significantly greater amounts of total Se; and (6) RN had the significantly greatest concentration of total Zn than the rest of the sites. SG and WH also had significantly greater concentrations.

**Table 7.** Mean dissolved and total metals concentration values for each site and standard deviation. Results are reported in μg/L.

| | SG | RN | 5P | D8 | CK | LG | WG | DW | SD | WH |
|---|---|---|---|---|---|---|---|---|---|---|
| As | 29 (±7.2) | 20 (±5.6) | 29 (±6.9) | 32 (±6.5) | 29 (±6.8) | 23 (±7.7) | 18 (±6.9) | 23 (±5.5) | 19 (±7.1) | 24 (±7.4) |
| As (T) | 34 (±6.8) | 25 (±10) | 31 (±7.4) | 34 (±8.3) | 32 (±7.9) | 26 (±7.3) | 21 (±4.0) | 25 (±5.5) | 23 (±7.2) | 29 (±8.1) |
| Cd | 0.20 (±0.17) | BDL | 0.20 (±0.13) | BDL | BDL | BDL | BDL | BDL | BDL | BDL |
| Cd (T) | 0.31 (±0.22) | 0.26 (±0.26) | 0.26 (±0.22) | 0.24 (±0.21) | 0.25 (±0.17) | 0.20 (±0.18) | 0.26 (±0.22) | 0.28 (±0.24) | 0.22 (±0.21) | BDL |
| Cu | BDL | 8.6 (±7.7) | 4.5 (±3.4) | BDL | BDL | 5.5 (±6.4) | BDL | 4.6 (±4.3) | BDL | 6.3 (±4.6) |
| Cu (T) | 9.0 (±19) | 21 (±30) | 7.9 (±11) | 6.8 (±10) | 11 (±16) | 7.4 (±12) | 6.9 (±6.6) | 5.6 (±5.3) | 7.1 (±7.6) | 6.8 (±5.6) |
| Cr | BDL | BDL | BDL | BDL | BDL | BDL | BDL | BDL | BDL | BDL |
| Cr (T) | BDL | 1.7 (±3.5) | BDL | BDL | BDL | BDL | BDL | BDL | BDL | BDL |
| Fe | 24 (±21) | 26 (±22) | 11 (±18) | 7.0 (±5.6) | 9.0 (±7.5) | 10 (±13) | 29 (±35) | 3.4 (±3.1) | 8.3 (±5.8) | 35 (±83) |
| Fe (T) | 90 (±47) | 516 (±1644) | 24 (±15) | 20 (±16) | 135 (±66) | 146 (±105) | 216 (±100) | 22 (±30) | 239 (±218) | 242 (±228) |
| Mn | 194 (±210) | 8.1 (±16) | 5.9 (±7.6) | 163 (±29) | 27 (±15) | 23 (±16) | 59 (±91) | 0.8 (±1.5) | 17 (±9.5) | 161 (±256) |
| Mn (T) | 196 (±205) | 30 (±82) | 5.6 (±7.4) | 163 (±37) | 54 (±20) | 36 (±14) | 85 (±92) | 1.3 (±1.9) | 35 (±21) | 171 (±258) |
| Ni | 4.6 (±1.4) | 3.2 (±1.2) | 3.0 (±1.3) | 3.8 (±1.0) | 3.9 (±1.3) | 3.1 (±1.3) | 3.2 (±1.3) | 3.1 (±1.0) | 3.1 (±1.0) | 4.8 (±1.5) |
| Ni (T) | 5.6 (±1.8) | 4.8 (±3.1) | 4.0 (±2.0) | 5.0 (±1.8) | 5.1 (±1.9) | 4.1 (±1.8) | 4.4 (±1.7) | 4.0 (±1.7) | 3.9 (±1.8) | 6.0 (±1.8) |
| Pb | BDL | BDL | BDL | BDL | BDL | BDL | BDL | BDL | BDL | BDL |
| Pb (T) | BDL | BDL | BDL | BDL | BDL | BDL | BDL | BDL | BDL | BDL |
| Se | 20 (±7.8) | 18 (±6.7) | 21 (±8.2) | 23 (±4.3) | 21 (±7.2) | 17 (±7.6) | 17 (±8.3) | 31 (±6.6) | 16 (±8.3) | 20 (±9.0) |
| Se (T) | 24 (±7.6) | 15 (±7.5) | 21 (±6.2) | 24 (±7.0) | 19 (±7.6) | 17 (±8.6) | 15 (±6.7) | 28 (±7.3) | 16 (±7.0) | 22 (±7.0) |
| Zn | 6.5 (±8.2) | 38 (±104) | 8.5 (±9.1) | 6.7 (±11) | 4.0 (±6.9) | 9.4 (±20) | 3.3 (±7.4) | 5.3 (±8.5) | 3.3 (±8.3) | 6.5 (±9.3) |
| Zn (T) | 8.9 (±8.5) | 48 (±114) | 8.5 (±8.9) | 4.2 (±4.1) | 3.8 (±5.4) | 4.3 (±4.7) | 4.6 (±5.8) | 5.0 (±6.7) | 5.1 (±9.7) | 11 (±13) |

(T) denotes total concentration. All other analytes are dissolved.

## 4.3. Correlations Analysis

Table 8 shows Spearman's rho correlations between mean pollutant concentrations and urban characteristics. Only significant correlations were displayed, and pollutants with no significant correlations were omitted from this investigation. Nutrients had strong correlations to a number of urban characteristics. However, pathogens had no significant correlations, and the only metals with significant correlations were As and Se.

**Table 8.** Spearman's rho correlations between mean pollutant concentrations and urban characteristics.

| Urban Characteristic | Nitrate | P | P (T) | K | TOC | As | As (T) | Se |
|---|---|---|---|---|---|---|---|---|
| Development, Open Space [1] | **−0.66** | −0.38 | −0.37 | 0.42 | 0.42 | −0.10 | −0.04 | −0.43 |
| Development, Low-Intensity [1] | −0.45 | **−0.70** | *−0.62* | −0.39 | **0.73** | *−0.56* | *−0.59* | **−0.66** |
| Development, Medium-Intensity [1] | 0.03 | −0.13 | −0.12 | −0.24 | 0.44 | **−0.67** | **−0.68** | −0.26 |
| Development, High-Intensity [1] | **0.68** | *0.59* | *0.51* | 0.42 | **−0.79** | *0.60* | 0.49 | **0.68** |
| Imperviousness [1] | **0.78** | **0.65** | **0.64** | **0.68** | −0.49 | 0.38 | 0.22 | 0.44 |
| Tree Canopy Cover [1] | **−0.72** | −0.34 | −0.36 | −0.38 | 0.39 | −0.26 | −0.21 | *−0.55* |
| Turf/Irrigated Land [2] | −0.42 | **−0.69** | *−0.61* | −0.42 | **0.72** | *−0.59* | *−0.62* | **−0.68** |
| Multi-Unit Residential [3] | 0.35 | **0.72** | **0.73** | *0.60* | −0.38 | 0.35 | 0.33 | 0.16 |
| Single-Unit Residential [3] | −0.30 | **−0.68** | **−0.66** | −0.47 | **0.72** | **−0.65** | **−0.65** | −0.49 |
| Industrial [3] | **0.75** | 0.10 | 0.07 | 0.04 | −0.28 | 0.09 | 0.09 | 0.31 |
| MED Density [4] | **0.76** | *0.53* | 0.47 | *0.55* | - | - | - | - |

[1] NLCD19, [2] DRCOG, [3] Denver, Jefferson, and Arapahoe County Land Use, [4] CDOR Marijuana Enforcement Division. Bold values indicate 95% significance. Underlined and Italicized values indicate 90% significance. MED correlations only analyzed for nutrients.

Notable urban characteristic correlation results include: (1) Nutrients (nitrate, P, and K) were positively correlated to characteristics of a more developed urban watershed, including percent imperviousness, multi-unit residential land use, and high-intensity development land cover. Conversely, nutrients were negatively correlated to features of a less developed urban watershed, including tree canopy cover, single-unit residential land use, turf/irrigated surfaces, and low-intensity development; (2) MED sites were strongly correlated with nitrate and K; (3) TOC was positively correlated with characteristics of a less developed watershed and negatively correlated with characteristics of a more developed urban watershed; and (4) metals As and Se were positively correlated with characteristics of a more developed urban watershed, and were negatively correlated to characteristics of a less developed urban watershed.

Table S2 shows correlations between pollutants. Notable significant results include: (1) Nitrate was positively correlated with dissolved Se and negatively correlated with TOC; (2) total P was positively correlated with TSS. Dissolved and total P were also positively correlated with K and Zn; (3) dissolved K was positively correlated with P, Mn, and Zn; (4) dissolved and total as were positively correlated with Se, Ni, and Mn and was negatively correlated with *E. coli* and TOC; (5) total Fe was positively correlated with TSS. Dissolved and total Fe was positively correlated with TOC, Mn, and Zn and was negatively correlated with nitrate and Se; (6) dissolved and total Mn were positively correlated with

Ni; (7) dissolved and total Ni were correlated with Se and negatively correlated with coliforms; (8) dissolved and total Se were negatively correlated with TOC, TSS, and Zn; and (9) total Zn was positively correlated to coliforms and *E. coli*.

## 5. Discussion

### 5.1. Dry-Weather Flows

Among the surface gulches (SG, CK, LG, WG, SD, and WH), flowrates appeared to be distributed by size, where the largest watershed (CK) showed the largest mean flowrate and the smallest watershed (SG) experienced the lowest mean flowrate. Since these drainages were perennial, they were likely fed by groundwater. However, this correlation between watershed size and flowrate suggests that these watersheds were also influenced by surface runoffs such as nuisance flows. Larger watersheds would imply that more urban processes and hydrologic connections exist and would contribute to larger dry-weather urban flows.

Conversely, there was no clear relationship between watershed size and flowrate at storm drain outfall locations (RN, 5P, D8, and DW), evidencing that storm drains may be dominated by groundwater contributions. D8 was observed to have the largest flowrate among the outfalls, yet was the second smallest watershed; it was also one of the most impervious watersheds with the greatest proportion of urban characteristics. More densely urbanized characteristics may indicate that a site's sewer network was deeper and thus may be more likely to be altered by groundwater flows [82,83]. Figure S1 shows where known groundwater seeps affect the South Platte River, and RN, 5P, D8, and DW appear to be directly next to or near significant seeps, further indicating that these sites were likely impacted by groundwater.

Because groundwater flow was a prominent source of dry-weather urban flow, it is important to understand how anthropogenic influences on shallow groundwater encourages groundwater contributions to urban surface streams. Though urban areas are often thought to decrease groundwater recharge due to impervious surface cover [84], several studies have shown that urban recharge has actually elevated shallow groundwater tables in urban areas through increased infiltration of nuisance flows, irrigation water, stormwater, and leaking water or wastewater pipes in the subsurface [49,50]. This is especially apparent in arid and semi-arid urban climates, where large amounts of water are imported, increasing the volume of water able to infiltrate [48,50]. A study by Paschke [48] found that the net groundwater recharge to the shallow Denver alluvial aquifer was nearly double what it was pre-development, evidencing that Denver likely experiences an artificially higher shallow groundwater table, which probably increases groundwater contributions to dry-weather urban flows. Similarly, net groundwater recharge has been documented to have increased in other arid and semi-arid cities, including Austin, Texas [49,85] and Barcelona, Spain [86], among many others. It is also possible that polluted urban discharges (e.g., stormwater infiltration, infiltration through polluted soils) contaminates the shallow groundwater supply [87].

### 5.2. Nutrients

Correlations between nitrate and urban characteristics suggest that nitrate was more prevalent in more developed watersheds, which is counter intuitive. However, the sites where nitrate was significantly greater were all storm drain outfalls. As concluded earlier in Section 5.1, storm drain outfalls were determined to be significantly impacted by groundwater contributions, suggesting that instead of urban land uses directly contributing nitrate, groundwater inflows were likely producing the excess nitrate pollution. A study on groundwater quality in the Denver Basin by Musgrove et al. [56] found that high concentrations of nitrate exist in the shallow groundwater of the Denver Basin, reporting values between 0.6–24.2 mg/L, which constrains the range of mean values found in this study (0.71–13.2 mg/L) and suggests that nitrate in urban drool was predominantly derived from groundwater supplies. The authors of the same study also concluded through isotope analysis that the majority of nitrate in the subsurface was derived from infiltrated

inorganic fertilizers from outlying areas of the Denver Basin, indicating that the high amounts of nitrate found at storm drain outfalls were likely from infiltrated irrigation flows containing nitrate-based fertilizers. Another line of evidence for groundwater derived nitrate include its correlations with dissolved Se. Se is naturally occurring in Denver's local geology [56], and it has been documented that nitrate is often strongly correlated with Se in groundwater because nitrate acts as an electron acceptor that mobilizes Se [88,89]. Therefore, sites with elevated nitrate and Se concentrations were likely groundwater driven. Furthermore, Figure S1 shows that the sites with the greatest concentrations of nitrate were near groundwater seeps, further evidencing that groundwater was the predominant source of nitrate.

Legacy sources of nitrate in the subsurface may also be a potential source of this contaminant in groundwater as Denver expands over formerly agricultural land uses [60–62]. The DDPHE also reported that in many of Denver's urban streams, N concentrations were greater during low-flow conditions, further suggesting that dry-weather nitrate pollution was groundwater dominated [35]. Overall, the high nitrate concentrations in Denver groundwater were likely sourced from pre-development and/or upgradient agriculture.

While there are many lines of evidence that suggest nitrate pollution was contributed by groundwater, other sources cannot be ignored. Nutrient runoff from MED sites within the city represent a potential source of nitrate pollution. MED site density and nitrate were positively correlated, which may imply that MED contributes to urban nitrate pollution. While additional targeted research would be helpful to confirm this conclusion, an interview with an anonymous source suggested that MED holders normally cultivate and dispose used nitrate-rich soil media on backyards (where they grow other plants) [90]. This agrees with MED waste management regulations and best management practices presented by the DDPHE, which recommends that MED waste is composted, encouraging at-home growers to start composting their nutrient-rich MED waste [91]. These backyard pollutants and composting are less likely to be mobilized via overland flow into an urban waterway during a storm but may infiltrate into shallow groundwater and eventually move to urban surface waterways during dry periods. Nitrate and nutrients from residential fertilizer application (i.e., lawns, ornamental vegetation, etc.), however, was not considered a dominant source of nitrate and nutrient pollution since nutrients and residential/less developed watersheds were significantly negatively correlated.

Atmospheric deposition was also a possible source of nitrate. Nitrous oxide emissions from industrial practices can potentially deposit in nearby urban streams [92,93], which could explain the strong positive correlation between nitrate and industrial land use. It is also important to note that a number of urban characteristics augment the concentration of nitrate in urban environments. Notably, poor nitrogen cycling due to impervious urban surfaces affects nitrogen retention and cycling, allowing nitrate to accumulate and persist in the urban environment [93,94], further emphasizing the prevalence and importance of nitrate as a pollutant of concern in dry-weather urban flows.

P was another pollutant of significant concern due to its exceedance of CDPHE standards. Given that P had many of the same previously explained correlations and patterns as nitrate, P may also be significantly loaded from groundwater contributions. P loading via groundwater is controversial since P was long believed to be immobile in the subsurface due to sorption or chemical precipitation [95]. However, P becomes mobile when in excess and soils and other ligands are no longer able to sorb additional P [95–97]. In addition, the typically reducing conditions in polluted urban groundwater facilitate P transport in the subsurface [98,99], suggesting that this nutrient exists and is able to be transported with groundwater flows. Similar to nitrate, P was also significantly greatest at storm drain sites, where groundwater seeps also existed (Figure S1), further indicating that P was likely derived from groundwater sources. The DDPHE also reported that in many of Denver's urban streams, P concentrations were greater during low-flow conditions, further indicating that dry-weather P pollution was groundwater dominated [35]. P is a common

nutrient in fertilizer, so groundwater derived P may have been sourced from upgradient agricultural activities.

Of course, other sources of P exist that may be contributing to urban pollution. Several studies pointed to increased impervious surface as facilitating P transport from decomposing vegetative litter to urban surface water. Since P was positively correlated with impervious area, it may be possible that accumulated decomposing vegetative litter on impervious drainage may be contributing additional P pollution in urban drool in Denver [45,46]. Furthermore, a characteristic of urbanization in semi-arid climates was turning ephemeral or intermittent streams perennial. Perennial flows increase streambank erosion, which may be P-bearing. Erosion created by construction practices could also introduce P-bearing soil into urban flows [100]. Active construction was observed in the RN watershed [101,102], and during times of observed active construction nearby the RN site, TSS and total P significantly increased. These results were evidence that construction practices may contribute to P pollution, though additional targeted research would be helpful to verify this conclusion. Additionally, a study from Monell et al. [103] found that Denver had natural concentrations of P within its soils, and polluted/disturbed soils had even greater concentrations of this contaminant, further suggesting that increased total P under conditions of higher TSS was likely from eroded Denver soils. Because P is a nutrient, it was also likely associated with MED sites, which is supported by a positive correlation between P concentration and MED site density. However, P is more strongly adsorbed to soils than nitrate or K, which means that it is less mobile in groundwater and may explain why correlations between MED and P were not as strong as nitrate or K. Given the variety of possible sources of P, more research is needed to determine the origin of P in dry-weather flows.

K is generally not studied as a predominant nutrient in literature as it is not acutely toxic and is typically not a limiting nutrient. However, significant correlations and high concentrations of K found in this study indicated that it may be relevant to urban water quality. Similar to nitrate and P, K was correlated to MED sites, potentially indicating that fertilizers used for *Cannabis* cultivation contributed greater K concentrations. K was also strongly correlated with P at all sites, suggesting that K and P may come from the same source, which is logical given both are components of fertilizer.

Groundwater should also be considered as a potential source of K. Musgrove et al. [56] found that shallow groundwater concentrations of K in Denver ranged from 0.79–12.9 mg/L, which was similar to the range of mean K concentrations found in this study (1.80–10.9 mg/L). K was greatest at D8, which was concluded to be significantly affected by groundwater, further implying that K may have derived from groundwater sources. MED infiltration and/or fertilizer infiltration from outlying agricultural areas may be sources of K in groundwater [104]. However, given a lack of prior studies, further targeted research may be needed to determine the specific sources of K in urban drool.

### 5.3. Pathogens

All sites exceeded CDPHE's surface water quality standard for *E. coli*, so it could reasonably be concluded that every analyzed urban site suffered from *E. coli* contamination, regardless of the urban characteristics within the watershed. The DDPHE also identified *E. coli* as being a consistent water quality concern in urban streams with Denver [35], agreeing with this study's assessment. For this reason, methods to mitigate widespread *E. coli* contamination are recommended.

Urban areas are environments of concentrated humans and urban wildlife (e.g., bats, squirrels, and raccoons), which increases the potential sources of bacteria from fecal matter. A study by Wu et al. [105] found that under certain circumstances urban wildlife may be a significant contributor of *E. coli*. However, leaks from human-derived sewage were also identified as a prevalent source of urban *E. coli* [14,20]. Further research is needed to determine the predominant sources of *E. coli* in Denver.

*5.4. Metals*

Though some sites experienced greater concentrations of As, the CDPHE chronic standard for total As was exceeded at every analyzed site, indicating that As was a contaminant of concern for every watershed. Correlations analysis suggests that prominent sources of As came from more developed and urban areas. However, As is primarily known as a groundwater contaminant due to its presence in the subsurface, especially in the Denver Basin, where urban groundwater concentrations were found in previous studies to reach up to 87.5 μg/L [56]. D8 had the highest mean total As concentration (34 μg/L), which was significantly greater than most sites. Since D8 was found to have significant groundwater contributions, this indicated that As was likely from groundwater sources rather than directly from urban land uses. Rising urban shallow groundwater tables and subsequent infiltration into urban waterways likely contributed to the presence of As in urban drool. In addition, increased urban recharge through As-bearing aquifers has been documented to mobilize As [106], which may explain why more developed watersheds of Denver experience greater As contamination. Dissolved and total As were positively correlated with Se, Mn, P, and K, which were also identified as potential subsurface contaminants, further suggesting that As may have been predominantly loaded from groundwater sources.

Though sources of As in the subsurface are often natural, As is also known to be occur in pesticides and wood preservatives, suggesting that further As contamination may also be from infiltration from outlying areas [58,107]. These results indicate that As in urban dry-weather flow was largely derived from natural geologic groundwater sources, but upgradient anthropogenic inputs may also represent a portion of As contamination. Further research may be needed to determine the specific sources of As in Denver's dry-weather flows.

Total Fe was not found to be significantly different among sites, but it had several very strong correlations with other pollutants, including TSS. This positive correlation with TSS suggests that mobilized soil particles contain significant amounts of Fe. This was especially prevalent at RN, where highly turbid samples, presumably from construction in the watershed, resulted in high concentrations of total Fe. This correlation was consistent among sites, evidencing the ubiquity of Fe in Denver soils. High concentrations of Fe were observed in the Denver subsurface in previous studies, validating these results [56]. In addition, the higher concentration of total Fe (opposed to dissolved Fe) indicated that Fe was mostly in its particulate form. The only exceedances of the total Fe-CDPHE standards were outliers, particularly when RN experienced increased TSS. Therefore, while Fe pollution may not be particularly concerning, this indicated that Fe-bearing soils were prevalent in Denver, which may hold implications for metal speciation.

Dissolved and total Mn was prevalent at all sites, with most of them exceeding the CDPHE chronic standard for dissolved Mn. SG, D8, and WH had significantly greater concentrations of Mn than the rest of the sites; since these three sites were relatively different from one another in terms of land use, there were no significant correlations with urban characteristics. Mn occurs naturally and is widely abundant in subsurface and surface flows [107,108]. However, anthropogenic sources were also found to exist in the urban environment. For example, a study by Tume et al. [109] found that Mn was contributed to urban soils from industrial inputs. WH had a significant amount of industrial land use nearby its sampling location, so industrial inputs of Mn may explain why this site had significantly greater concentrations of this contaminant. Hou et al. [110] also found that Mn was greater in urban groundwaters than non-urbanized areas, suggesting that urban land uses and urbanization may increase Mn concentrations in urban surface water and groundwater (for the latter, likely via infiltration from surface pollution).

Conversely, Mn was also positively correlated with pollutants, such as As, that were geologic-bearing and naturally occurring in the soil and subsurface of the Denver basin, suggesting that dry-weather Mn may have been derived partly from natural sources. Musgrove et al. [56] reported a very wide range of Mn in urban shallow groundwater in Denver, ranging from 0.24 to 663 μg/L, which includes the means values found in this

study (1.3–196 µg/L). Evidence supports that Mn may have been geologically derived, but anthropogenic inputs and effects likely contributed to Mn pollution as well; however, a lack of correlations with urban characteristics and high concentrations present in three very different watersheds complicate this assessment.

Ni may come from a variety of sources, including anthropogenic pollution and natural sources [107,109]. Because Ni is correlated with other pollutants that were associated with soil and groundwater (As, Mn, and Se), it may be likely that Ni was sourced from naturally forming geologic sources. Reported values of Ni in this study (3.9–6.0 µg/L) fit within the range of values of Ni in Denver's urban groundwater found by Musgrove et al. [56] (0.06–19.7 µg/L), which supports that Ni may have been loaded by groundwater contributions. However, Ni is a well-known anthropogenic pollutant that accumulates in soil from traffic pollution, industrial inputs, and leaching metallic infrastructure [109]. Davis et al. [107] found that Ni is usually contributed by both natural and anthropogenic sources in urban soils, which may indicate why Ni was present at all sites, but WH, which had a notably large portion of industrial land use towards its sampling point, had the greatest mean concentration of Ni. Higher concentrations in a catchment such as WH that contains industrial sites are logical, assuming larger commercial vehicles are frequently stopping and starting, causing brake and engine parts wear, a common source of Ni pollution. WH drainage is well connected to the surface activities (i.e., has traditional catchment topography, albeit urbanized, but not a storm sewer outfall). WH also exceeded chronic CDPHE standards for Ni at times, marking it as a potential contaminant of concern in dry-weather flows.

Though Se was positively correlated to highly developed watersheds, many lines of evidence suggest that Se derived from groundwater sources rather than direct urban surface inputs. The Denver Basin was documented to contain high concentrations of Se in the subsurface, where Se concentration reached as high as 696 µg/L in shallow groundwater below the city [56]. Se was also positively correlated with other pollutants associated with soil and groundwater sources (nitrate and As), supporting the theory that Se was linked to shallow groundwater sources. Additionally, Se was greatest at sites where groundwater was determined to be major contributors of dry-weather flow (Figure S1), also suggesting that the majority of Se was coming from subsurface geology. The DDPHE also reported that Se concentrations were greater under conditions of low-flow in Denver, further proving that Se was groundwater driven [35].

Though Se is recognized mainly as a natural pollutant, anthropogenic activities have been documented to promote further mobilization of Se to urban water bodies, including, but not limited to, irrigation practices, rising urban groundwater tables, and urban recharge through Se-bearing geology [33,58,106,111]. A specific example from a Southern California watershed studied by Hibbs et al. [63] found that urban development drained and oxidized formerly reduced marshland sediments, mobilizing accumulated Se, which significantly polluted receiving urban water bodies. Due to the combination of natural Se deposits in the Denver basin and anthropogenic influences that encourage Se-polluted waters from the subsurface to enter surface streams, Se concentrations in Denver were considerably high. Due to the acute exceedance of CDPHE standards and ubiquity among sites, Se should be further evaluated and addressed for its environmental consequences.

Though Zn had no significant correlations to urban characteristics, correlations to P and K suggested that Zn may have been associated with fertilizers from MED sites and/or infiltrated from outlying agricultural areas. Zn is often associated with fertilizers as it is an essential plant micronutrient. Total Zn ranged from 3.8 to 11 µg/L among sites (with the exception RN), which was within the range of Zn typically found in natural waters and shallow groundwater in the Denver basin (0.6–19.9 µg/L) [56]. The similar concentrations of Zn among sites suggested that natural geologic sources were its primary source in urban drool. However, the only derivation from this conclusion was the results from RN. Several samples collected from RN had extremely high TSS, presumably from construction within the watershed. Under these conditions of high TSS, total Zn increased

significantly, indicating that Denver's sediments had large quantities of Zn built up from urban processes and may contribute to urban drool pollution. These urban sources of total Zn are usually derived from traffic pollution and urban building materials; two studies demonstrated that the highest concentration of heavy metals such as particulate Zn were typically found near roads and galvanized building materials [112,113]. Due to the variety of possible sources of Zn, further targeted research is recommended to determine the specific sources of Zn in Denver.

### 5.5. Relationship between TSS, P, and Metals

RN events collected on 6 April and 28 May (2021) had extremely high concentrations of TSS, which also resulted in high concentrations of total P and total metals, including As, Cu, Fe, Mn, Ni, Pb, and Zn (Figure S2). This enhanced nutrient and metal pollution was perceived to be from contaminated urban soils, since pollutants were largely in particulate forms. Several studies have identified that heavy metals associated with urban processes such as vehicular use, urban building materials, and industrial land uses accumulate over time in urban soils [53,112,113]. Similarly, naturally-occurring and fertilizer-derived P can accumulate in urban soils over time [100,114]. Stormwater runoff, construction, or other erosion-causing events can disturb and mobilize these soils into urban waterways.

The cause of higher TSS under dry-weather conditions was determined to be construction due to the documented activity that occurred in the RN watershed during the time of collection [101,102]. Notably, construction included significant road work, which may explain why vehicular-based pollutants, including Cu, Pb, and Zn, were prevalent. In addition, though construction had been present in the watershed since early 2021, the only occurrence of extremely high TSS occurred during weekday collections, when construction activities were underway; high TSS was not observed during weekend collections when no construction activities were actively underway. This further indicated that construction activity was the cause of high TSS at RN and also highlights the rapid feedback between construction activities and urban water pollution. A number of studies have also identified that construction and development practices can mobilize sediment and various pollutant in the urban environment [115,116]. Though other factors may be contributing to higher TSS, dry-weather results found in this study suggest that construction-disturbed urban soils were a source of increased total P and metal pollution.

### 5.6. Pollution Mitigation Strategies

Mitigation strategies to improve water quality are referred to as best management practices (BMPs). BMPs are widely used to reduce pollution from stormwater runoff, and the effective use of BMPs has been documented to greatly improve water quality. However, dry-weather mitigation efforts will likely look and function differently than stormwater BMPs, which traditionally focus on end-of-pipe solutions, retention, and detention. While stormwater BMPs may provide some sort of dry-weather pollution mitigation, dry-weather pollution would be best managed through in-stream treatment, diversion and treatment (either engineered or green infrastructure), and/or source control [9]. Findings from this study revealed important relationships between urban catchment characteristics and water quality parameters that can inform dry-weather BMPs.

### 5.7. Future Research

Future research can help us to better understand the dynamics of urban dry-weather water quality. Given the results of this study, suggestions for future research are listed below:

- Even though sampling for this study lasted a year, a longer study would be more indicative of water quality since polluting factors and land use in the urban environment are constantly changing [37]. Better characterization can be made by taking more frequent samples over a 3–5 year time period.
- Because much of the dry-weather urban water pollution was implied to derive from groundwater sources, it would be beneficial to characterize urban shallow groundwa-

ter quality, study the connections between shallow groundwater and urban surface water, and evaluate how the urban environment affects shallow groundwater table behavior, such as identifying sources of infiltration/percolation.

- Though many pollutants were attributed to shallow groundwater sources, further research is recommended to determine the origin of pollutants in Denver, specifically P, K, *E. coli*, As, Mn, Ni, and Zn.
- Strong correlations between MED sites and fertilizer components nitrate, P, and K suggest that targeted research should be performed to confirm the impact of MED sites on urban water quality and to inform potential mitigation. There appears to be no limit on the number of approved MED sites, suggesting that MED sites may continue to grow, suggesting urgent study is needed.
- Additional targeted investigation into the instances of high TSS and erosion, including the effects of construction and development, is recommended. A better spatial characterization of Denver soils for target pollutants is also recommended.
- Several studies indicated that there are beneficial reuse strategies for dry-weather urban flows, such as for irrigation [8,117]. As water scarcity increases in Denver, it may be important to evaluate the reuse potential of dry-weather flows.
- Dry-weather flow water quality is likely an indication of shallow urban groundwater quality. Shallow urban groundwater is not a common water resource for cities in any location because it is typically presumed to be polluted. However, as water shortages grow, particularly in the arid western U.S., cities will likely need to utilize shallow groundwater as municipal water supply. For these reasons, more extensive and frequent monitoring of urban drool is recommended.

## 6. Conclusions

Pollution was prevalent in urban drool in Denver; this pollution negatively affects the South Platte River, which is one of Denver's primary drinking-water and recreational resources. Dry-weather flows were found to be influenced considerably by groundwater inflows, which drives the perennial flow. Storm sewer drains were determined to be substantially affected by groundwater contributions due to their consistency in flow, depth/proximity to groundwater tables, and location near known groundwater seeps along the South Platte River. Most contaminants in dry-weather flow were found to be loaded at least partially by shallow groundwater sources. Urban development impacts on the shallow groundwater table most likely augmented groundwater contributions and contamination to dry-weather flows; this likely occurred through enhanced urban recharge. Other important conclusions on dry-weather water quality from this study include the following:

- Pollutants of concern as identified by exceedances of CDPHE standards in at least one watershed included nitrate, total P, *E. coli*, As, Mn, Ni, and Se. Nitrate and Se were of particular concern due to their exceedances of acute standards. *E. coli* was also of special concern due to its ubiquitous contamination throughout Denver, regardless of the watershed's characteristics.
- Evidence suggested that nitrate, Ni, and Mn may have been partially contributed by industrial inputs and potentially commercial traffic.
- MED sites may have been a significant contributor of nitrate, P, K, and Zn.
- Instances of high TSS were notable due to severe contamination of particulate pollutants, including P and metals. These particulate pollutants were attributed to contaminated urban soils and were presumed to have been contributed via erosion during construction in the watershed, though other erosive activities may have contributed to enhanced TSS.

Conclusions from this study revealed important relationships between urban catchment characteristics and dry-weather surface water quality that can inform BMP design. Increasing urbanization and predicted drier climates due to global warming suggest that dry-weather urban surface water will likely become more important to properly maintain

and manage. The results from this study provide insight on dry-weather water quality and urban water management for the City of Denver and similar semi-arid urban areas.

**Supplementary Materials:** The following are available online at https://www.mdpi.com/article/10.3390/w13233436/s1, Figure S1: Map of groundwater seeps along the South Platte River. Table S1: One-factor permutation test for difference in means results. Table S2: Spearman's Rho correlations between selected analytes. Figure S2: RN under conditions of high concentrations of TSS.

**Author Contributions:** Conceptualization, J.E.M.; methodology, F.G.P., P.A.G.-C. and J.E.M.; validation, P.A.G.-C. and J.E.M.; formal analysis, F.G.P. and J.E.M.; writing—original draft preparation, F.G.P.; writing—review and editing, P.A.G.-C. and J.E.M.; visualization, J.E.M.; supervision, P.A.G.-C. and J.E.M.; project administration, P.A.G.-C. and J.E.M.; funding acquisition, J.E.M. All authors have read and agreed to the published version of the manuscript.

**Funding:** This work was supported by the National Science Foundation-funded Engineering Research Center (ERC) for Reinventing the Nation's Urban Water Infrastructure (ReNUWIt) (NSF EEC-1028968), a grant from the Colorado Higher Education Competitive Research Authority. Additional funding was also provided by the Edna Bailey Sussman Foundation.

**Data Availability Statement:** The data presented in this study are available in "Urban Drool Water Quality in Denver, Colorado: Pollutant Occurrences and Sources in Dry-Weather Flows" and in the Supplementary Materials.

**Acknowledgments:** We gratefully acknowledge the City and County of Denver and the City of Golden for providing data and their assistance.

**Conflicts of Interest:** The authors declare no conflict of interest. The funders had no role in the design of the study; in the collection, analyses, or interpretation of data; in the writing of the manuscript, or in the decision to publish the results.

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
