# Peer review of "Urban Drool Water Quality in Denver, Colorado: Pollutant Occurrences and Sources in Dry-Weather Flows"

_water, doi:10.3390/w13233436_

Round 1
Reviewer 1 Report
The manuscript water 14533469 entitled " Urban Drool Water Quality in Denver, Colorado: Pollutant Occurrences and Sources in Dry-Weather Flows)", where the authors analyze the different sources of urban drool pollution in Denver, Colorado by identifying relationships between urban catchment characteristics and pollutants.
Although the experimental design is adequate for the purposes of the study, the manuscript needs a major revision. I have any suggestions such as:
The introduction provides extensive information regarding the role of urban slime in and around the city of Denver. However, the authors should condense a little more since there is data that is repeated in different sentences of the text.
Materials and methods
- Why is it necessary to rinse the bottle and its cap before filling the bottles with urban slime?
- Why do the authors in some cases indicate only the chemical symbol of the metal and in others give the name and the symbol? I think that the entire scientific community knows the periodic table of chemical elements.
- In the statistical analysis section, the authors also give a lot of information that we all already know and that it is not necessary to explain (for example, meaning of a +1 or -1 correlation). Please remove unnecessary explanations.
- The authors indicate that many of the variables analyzed do not follow a normal distribution, so a non-parametric analysis had to be carried out. Have the authors considered the possibility of some kind of mathematical transformation of these variables? Many times, this transformation makes the new variable follow a normal distribution, facilitating subsequent statistical analysis.
Results and Discussion
Table 2 should be listed as table 1 since it is the first one named in the text (line 210, subsection 3.1). In addition, subsection 4.1 of results does not truly correspond to results and should be included in subsection 3.1. Study Area and Site Selection.
Lines 409-412 of sub-section 4.3 are clearly materials and methods that should be included in sub-section 3.3. Field Collection.
In addition, the authors indicate "Boxplots are shown for contaminants that exceeded CDPHE standards" (lines 412-418), but in what figure?
Regarding the discussion, the authors must be clearer and more concise in their arguments, which are repeated everywhere. The discussion should not include results, nor refer to the figures present in the text or to those of the supplementary material. In addition, there are phrases that are clearly results, that on the other hand, are not reflected in their corresponding section or that were not detailed enough. Therefore, I consider that a new rewriting of the entire discussion section is necessary.
On the other hand, the conclusions should be short and well defined the opposite of what the authors have done.
As for future research, it should go to the end of the discussion.
Author Response
We thank Reviewer 1 for her/his comments and suggestions. Please see the attachment (Reviewer 1 Response water-1453469) for point-by-point responses.

Reviewer 2 Report
In my opinion, the paper can be accepted in this present form and can be ready for publication.
Author Response
We thank Reviewer 2 and appreciate her/his time and review.
Round 2
Reviewer 1 Report
The authors have greatly improved the presentation of the materials and methods sections as well as the results and discussion. I congratulate them for the immense research work done and encourage them to carry out the future research they have proposed at the end of the manuscript.